# Neural mechanisms of social learning in the female mouse

Yuan Gao[1], Carl Budlong[1], Emily Durlacher[2], Ian G Davison[1]*

[1]Department of Biology, Boston University, Boston, United States; [2]Program in Neuroscience and Behavior, Mount Holyoke College, South Hadley, United States

**Abstract** Social interactions are often powerful drivers of learning. In female mice, mating creates a long-lasting sensory memory for the pheromones of the stud male that alters neuroendocrine responses to his chemosignals for many weeks. The cellular and synaptic correlates of pheromonal learning, however, remain unclear. We examined local circuit changes in the accessory olfactory bulb (AOB) using targeted ex vivo recordings of mating-activated neurons tagged with a fluorescent reporter. Imprinting led to striking plasticity in the intrinsic membrane excitability of projection neurons (mitral cells, MCs) that dramatically curtailed their responsiveness, suggesting a novel cellular substrate for pheromonal learning. Plasticity was selectively expressed in the MC ensembles activated by the stud male, consistent with formation of memories for specific individuals. Finally, MC excitability gained atypical activity-dependence whose slow dynamics strongly attenuated firing on timescales of several minutes. This unusual form of AOB plasticity may act to filter sustained or repetitive sensory signals.

## Introduction

Chemical cues detected by the vomeronasal system convey vital social information, influencing diverse behaviors such as reproduction (*Bruce and Parrott, 1960*; *Kimchi et al., 2007*), pair bonding (*Young and Wang, 2004*), parental care (*Dulac et al., 2014*; *Kendrick et al., 1992*; *Lévy et al., 2004*), individual recognition (*Hurst, 2009*), and aggression (*Chamero et al., 2007*; *Stowers et al., 2002*). Vomeronasal pathways directly access the limbic system, consistent with their powerful role in guiding behavior, and also influence neuroendocrine centers to modify physiological and hormonal status (*Dulac and Torello, 2003*; *Tirindelli et al., 2009*).

While vomeronasal circuits often elicit stereotyped behavioral and neuroendocrine responses, they can also be highly plastic. In one striking example, female mice imprint on the pheromones of the stud male after mating, where a single salient sensory experience drives long-term changes in both behavior and the flow of sensory information to central targets (*Keverne and Brennan, 1996*). During the first few days after fertilization, chemosignals from unfamiliar males typically block pregnancy by altering the female's neuroendocrine state (*Bruce and Parrott, 1960*). However, mating opens a plasticity window that creates a recognition memory for the stud's pheromones, so that they lose their potency and no longer disrupt embryo implantation (*Brennan and Keverne, 1997*). Memories are formed within hours, yet last weeks or longer (*Kaba and Keverne, 1988*). Sensory imprinting thus offers an opportunity to test the neural basis of a natural form of social learning in a circuit intimately coupled with intraspecies behaviors.

While social experience acts on diverse neural circuits throughout the brain (*Wallace et al., 2009*; *Wu et al., 2014*), mating-dependent learning is strongly linked to plasticity in the accessory olfactory bulb (AOB). Imprinting in females leads to local neurochemical changes (*Brennan et al., 1995*), is affected by local lesions or pharmacological interventions (*Brennan and Keverne, 1997*; *Kaba et al., 1994*; *Kaba and Keverne, 1988*), and can be artificially induced by manipulating AOB

*For correspondence: idavison@bu.edu

**Competing interests:** The authors declare that no competing interests exist.

**eLife digest** To navigate social situations, humans and other animals need to remember who they have interacted with and how it went, and adjust their behavior in future encounters accordingly. For example, your physical actions, and even your body's physiological responses, will be very different when you encounter the last person you kissed instead of the last person you fought with (assuming this is not the same person!).

Memories of social interactions can have dramatic consequences. For instance, male mice often kill the offspring of other males. Female mice appear to have adopted a countermeasure to avoid losing a litter of pups to such aggression: they will spontaneously abort a pregnancy when exposed to chemicals called pheromones from unfamiliar males. However, when the female mouse is exposed to the pheromones of the male she mated with she maintains her pregnancy. Exactly how the memories of previous social interactions with the males affect the female's pheromone responses is not fully understood.

To investigate how the female is able to respond differently to different males, Gao et al. recorded the activity of individual neurons taken from the brain tissue of female mice who had recently mated. The recordings showed that previous social experiences produce learning-related changes in the brain of the female mouse that reduce how sensitively pheromone-detecting neurons respond to the chemical cues of the male mate. This suppresses the signals that the neurons would otherwise send to trigger an abortion in response to male pheromones.

Gao et al. also used fluorescent tags to identify which neurons in the female's brain had been activated during mating. This revealed that only those neurons that had been activated by the mate become unresponsive when the cells again encountered his pheromones. This suggests that a set of neurons in the female's brain records the chemical 'fingerprint' of the mate, and can then selectively filter out that mate's pheromone signals.

Many other social interactions, such as parenting, are also strongly shaped by experience. The results presented by Gao et al. may therefore offer wider lessons for understanding how the brain targets different behaviors toward specific individuals. It will also be important to investigate how highly arousing experiences cause such powerful memories to form. This could ultimately help us to better understand – and potentially treat – conditions like post-traumatic stress disorder.

signaling (*Kaba et al., 1994*). Pheromonal cues are encoded in AOB by the firing of mitral cells (MCs), whose activity signals gender, hormonal status, and in particular, strain and/or individual identity (*Ben-Shaul et al., 2010*; *Luo et al., 2003*; *Tolokh et al., 2013*). Well-established theories propose that learning selectively suppresses the firing of the MCs encoding the stud male so that his pheromones no longer drive neuroendocrine responses, precluding pregnancy block (*Brennan, 2004*; *Brennan et al., 1990*). MC suppression is further proposed to depend on strengthening of local inhibitory circuits in AOB, consisting largely of granule cells (GCs) that supply feedback inhibition to MCs through unique dendrodendritic synapses (*Isaacson and Strowbridge, 1998*; *Shepherd and Greer, 1998*). Inhibitory plasticity is consistent with both microdialysis data (*Brennan and Binns, 2005*; *Brennan et al., 1995*) and ultrastructural changes in local interneurons (*Matsuoka et al., 1997*, *2004*).

Despite well-established models of learning in AOB, many key features of plasticity remain untested. To date, direct measurements of either synaptic plasticity or changes in MC output are lacking. Furthermore, while the selectivity of recognition memories for different individual or strains is thought to rely on changes in specific groups of MC, there are no data linking plasticity to functionally defined cell populations. More broadly, the nature of the neural changes that allow for adaptive changes in social behavior remain poorly understood.

Here, we examined how mating affects local AOB microcircuits using targeted whole-cell recordings of identified neurons activated by the stud male in ex vivo brain slices. We found pronounced reductions in the sensitivity of AOB neurons, which unexpectedly were mediated by changes in intrinsic excitability rather than synaptic strength, suggesting a novel cellular basis for encoding sensory memories in AOB. MC firing was selectively attenuated in stud-activated neurons, suggesting a

potential basis for the specificity of pheromonal learning. Changes in MC responsiveness emerged only when they were activated with repetitive patterns, suggesting that after learning the AOB may dynamically filter repetitive sensory signals from the stud male, lessening their impact on neuroendocrine status on long timescales.

## Results

### Slow, powerful self-inhibition in AOB MCs

While inhibitory circuits are intensively studied in main olfactory bulb (*Isaacson and Strowbridge, 1998*; *Shepherd and Greer, 1998*), their role in shaping AOB output is not well characterized. We thus began by characterizing self-inhibition in AOB projection neurons, mitral cells (MCs). MC self-inhibition arises from specialized dendrodendritic synapses shared with local interneurons, primarily granule cells (GCs) (*Figure 1A*). We assessed self-inhibition by driving MC firing with current injection and examining the resulting synaptic feedback from interneurons. Brief, high-frequency spike

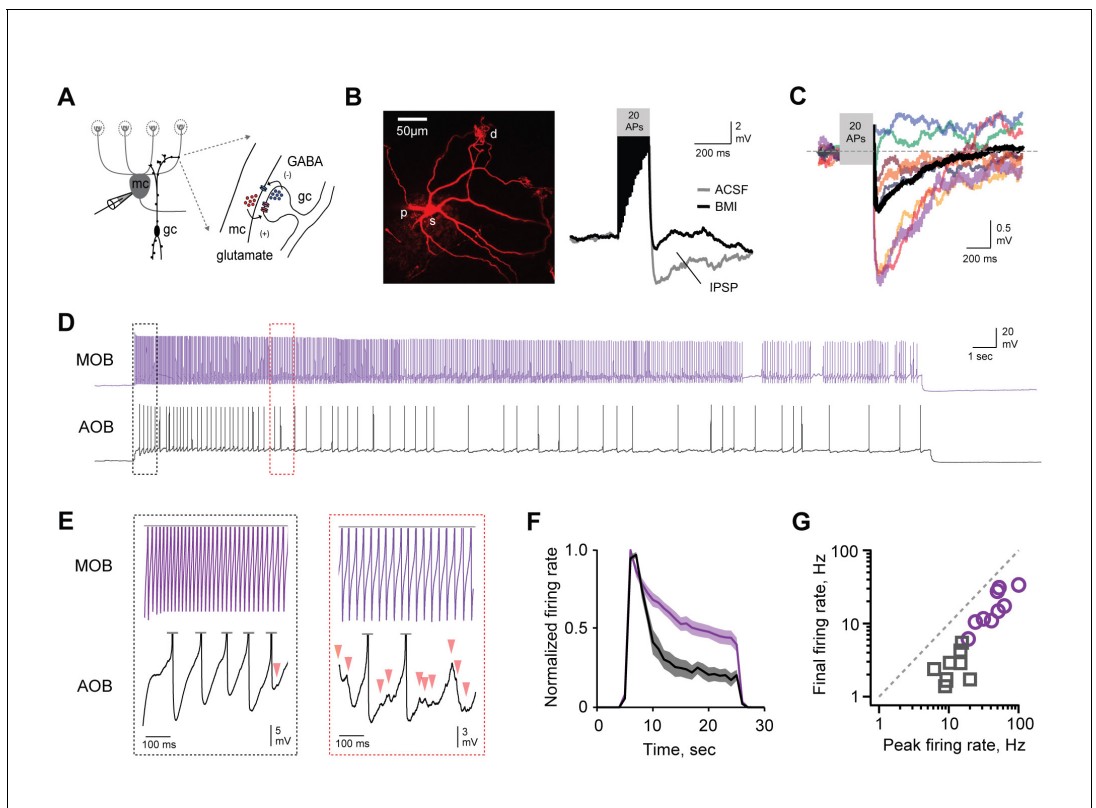

**Figure 1.** AOB MCs express robust, slowly emerging self-inhibition. (**A**) Schematic of dendrodendritic self-inhibition pathway in MCs. (**B**) Left, dye-filled MC imaged after recording. S, soma; d, dendritic tufts that integrate sensory inputs; p, recording pipette. Right, brief, high-frequency spike trains trigger modest self-inhibition (gray, standard ACSF; black, after blocking inhibition with 15 µM BMI; IPSP, inhibitory postsynaptic potential). (**C**) Pharmacologically isolated self-inhibition in AOB MCs. Colored traces show individual cells; black trace, average; mean $V_{inh}$ = −1.4 ± 0.27 mV (n = 8 cells in 5 mice). (**D,E**) Prolonged firing elicits robust MC self-inhibition in AOB but not MOB (black and purple respectively). Boxes show expanded view of barrages of IPSPs in AOB MCs, indicated by arrowheads, which only emerge after several seconds of firing. (**F**) Self-inhibition contributes to stronger decay of MC firing rates in AOB during extended stimuli (n = 9 and 9 cells in 5 and 5 mice for AOB and MOB respectively). (**G**) Initial and final firing rates during MC spike trains in MOB and AOB (purple and gray respectively).

The following source data and figure supplement are available for figure 1:

**Source data 1.** This spreadsheet contains the initial and final firing rates for the individual neurons shown in *Figure 1G*.

**Figure supplement 1.** Robust self-inhibition regulates spiking of AOB MCs.

trains generated only modest feedback inhibition in AOB (*Figure 1B–C*), comparable but smaller on average than reported in main olfactory bulb (*Abraham et al., 2010*; *Margrie et al., 2001*). However, pronounced inhibition appeared when we drove MCs with prolonged stimuli similar in duration to chemosensory responses, which can last for tens of seconds (*Luo et al., 2003*). Extended MC spike trains elicited slowly emerging but robust barrages of inhibitory postsynaptic potentials (IPSPs) that contributed to a strong decline in firing rate (*Figure 1D–1F*; *Figure 1—source data 1*). In contrast, the same protocol generated little detectable inhibition in MCs of the main olfactory bulb, where high firing rates were sustained throughout the train. To confirm that synaptic self-inhibition shapes MC output, we eliminated dendrodendritic feedback pathways by blocking fast synaptic transmission with NBQX, APV, and gabazine (5, 25, and 10 µM respectively). Pharmacologically eliminating feedback inhibition typically increased MC firing as well (*Figure 1—figure supplement 1*). Together, our results suggest that MC self-inhibition is substantially stronger in AOB than in main olfactory bulb, but also unusually slow to manifest, consistent with the prolonged sensory responses characteristic of this brain area. Such powerful self-inhibition by single MCs could potentially provide a basis for cell-specific control over AOB output, as previously proposed (*Brennan and Keverne, 1997*).

## Mating enhances synaptic inhibition in AOB

Microdialysis suggests increased bulk GABA release in AOB after mating (*Brennan and Binns, 2005*), consistent with enhanced inhibition, but the synaptic correlates of imprinting have not been measured directly. We next examined the effects of pheromonal learning on local inhibitory circuits. To align the timing of mating with brain slice recordings (*Figure 2A*), we induced estrus using ovariectomy, implanted estradiol capsules, and progesterone injection (*Ström et al., 2012*). We then paired females in their home cage with sexually experienced males for 4 hr to provide the mating and sensory exposure required for imprinting. Females engaged in frequent, repetitive investigation of males, particularly of facial and anogenital regions (*Figure 2—figure supplement 1A–C*; mean interval, 62.1 ± 7.99 s; median, 15.9 s). Investigative behavior was elevated in mated relative to sensory-exposed females, suggesting that they experienced both heightened arousal states and increased levels of sensory input during the pairing period (*Figure 2—figure supplement 1D*).

Immediately following mating and sensory exposure, we prepared AOB brain slices from females and examined changes in synaptic inhibition with whole-cell voltage-clamp recordings of MCs. We compared three groups: (i) mating plus sensory experience with a freely moving male; (ii) sensory-exposed controls without mating; and (iii) naïve mice with no prior male exposure, housed overnight in a fresh cage. We measured GABAergic input onto MCs by recording miniature inhibitory postsynaptic currents (mIPSCs; *Figure 2B*). IPSCs were pharmacologically isolated with 5 µM NBQX, 25 µM APV, and 1 µM TTX (*Figure 2—figure supplement 2*). The frequency of mIPSCs was strongly increased in mated animals vs. naïve and sensory-exposed groups (*Figure 2C–D*; 2.45 ± 0.37 Hz vs. 1.48 ± 0.21 Hz and 1.28 ± 0.17 Hz respectively). The mean amplitude of mIPSCs was similar for all three conditions, suggesting little change in the postsynaptic sensitivity of inhibitory synapses onto MCs (*Figure 2E–F*; 56.8 ± 3.5 pA, 54.5 ± 3.8 pA, and 65.8 ± 6.1 pA for naïve, sensory-exposed and mated mice respectively; *Figure 2—source data 1*). The distribution of amplitudes was shifted towards higher values, however, indicating that a subset of inhibitory synapses may be strengthened. Overall, mating experience substantially increased inhibitory input onto MCs, consistent with prior microdialysis results (*Brennan et al., 1995*).

Because imprinting may act on other elements of the self-inhibition pathway, we also asked whether mating alters excitatory input to granule cells, the most numerous interneurons in AOB. Using current clamp recordings from GCs, we measured the amplitude and frequency of spontaneous excitatory postsynaptic potentials (sEPSPs; *Figure 2G*). The frequency of sEPSPs was elevated in mated relative to naïve and sensory-exposed females (*Figure 2H–I*; 4.25 ± 0.71, 3.57 ± 0.56, and 6.61 ± 0.63 Hz for naïve, exposure and mated groups respectively). EPSP amplitude was also slightly enhanced in mated compared to naïve animals (*Figure 2J–K*, 1.08 ± 0.08, 1.18 ± 0.10, and 1.44 ± 0.10 mV for naïve, exposed, and mated groups respectively). These differences were consistent across a wide range of EPSP detection criteria (*Figure 2—figure supplement 3*). Overall, learning also increased excitatory drive onto GCs, enhancing both presynaptic and postsynaptic elements of glutamatergic synapses. Together, our data suggest that imprinting upregulates both the excitatory and inhibitory components of the pathways for MC self-inhibition.

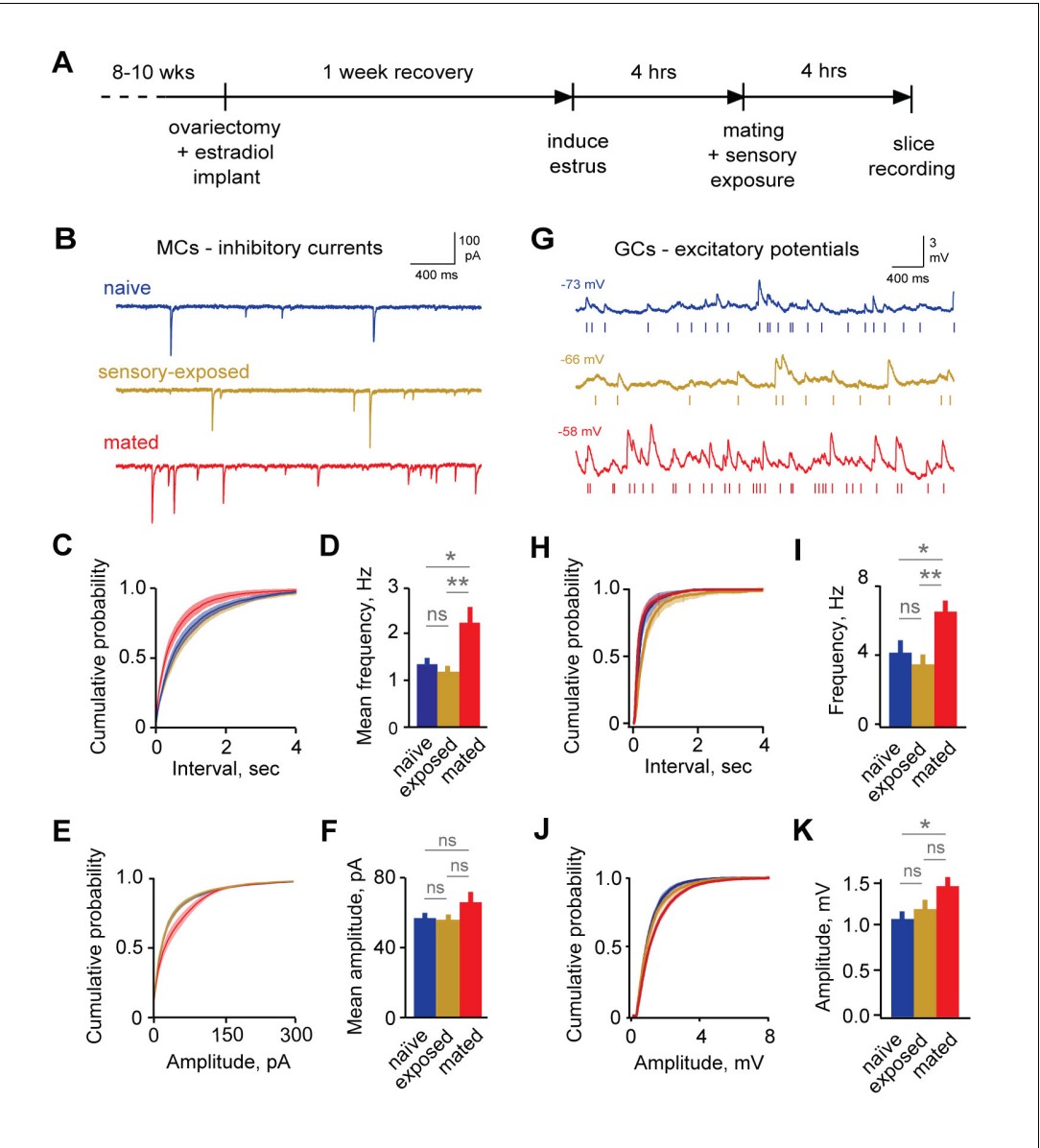

**Figure 2.** Imprinting drives synaptic plasticity in both MCs and GCs. (**A**) Schematic of timeline for mating, sensory experience, and recording. (**B**) Inhibitory synaptic inputs recorded in voltage-clamped MCs from naïve, sensory-exposed, and mated mice. (**C,D**) Mating substantially increases mIPSC frequency. Left, cumulative interval distributions; mated < naïve and exposed groups, p=0.002 and 0.001 respectively. Right, mean frequency (F = 5.88; Fc = 3.20; p=0.005 for mated vs. naïve; ANOVA with post-hoc Tukey test; n = 18, 17, and 15 cells in 5, 5, and 6 mice respectively). (**E,F**) The mean amplitude of mIPSCs was not significantly changed by imprinting (F = 1.74; Fc = 3.20; p=0.19; ANOVA with post-hoc Tukey test), although distributions were significantly shifted towards smaller values in the mated vs. naïve and sensory-exposed groups (p=0.00007 and $3 \times 10^{-7}$ respectively). (**G**) Example traces showing spontaneous EPSPs in GCs from naïve, sensory-exposed and mated mice. Rasters indicate synaptic events used for analysis. (**H,I**) Mating increased mean sEPSP frequency relative to both naïve and sensory-exposed animals (F = 6.64; Fc = 3.14; p=0.00037 and 0.038 for mated vs. exposed and naïve mice respectively; ANOVA with post hoc Tukey test; n = 17, 19, and 30 cells in 5, 9, and 12 mice). Interval distributions were significantly smaller for mated vs. exposed and naïve animals (p=$1 \times 10^{-11}$ and 0.0008 respectively). (**J,K**) Mating also increased mean sEPSP amplitude in mated vs. naïve animals. Left, cumulative distribution; right, mean amplitude (F = 3.56; Fc = 3.14; p=0.037 for naïve vs. mated, ANOVA with post hoc Tukey test). Amplitude distributions were larger for mated vs. naïve mice (p=0.04). NS, not significant; *p<0.05; **p<0.001.

*Figure 2 continued on next page*

*Figure 2 continued*

The following source data and figure supplements are available for figure 2:

**Source data 1.** This spreadsheet contains the mean frequency and amplitude data for the individual neurons used to generate the bar plots shown in *Figure 2D and F* (mitral cell mIPSCs) and 2I and 2K (granule cell mEPSCs).

**Figure supplement 1.** Mating and sensory interactions during pairing.

**Figure supplement 2.** Pharmacologically isolated inhibitory synaptic currents in MCs.

**Figure supplement 3.** Synaptic effects in GCs are independent of event detection criteria.

## Synaptic plasticity lacks cellular selectivity

Pheromonal recognition memories are specific to particular individuals or strains, implying that learning may act selectively on the particular AOB neurons activated by the stud's chemosignals (*Keverne and Brennan, 1996*). To test the cellular specificity of plasticity, we identified the AOB neurons activated during mating and sensory exposure using GFP reporter lines based on the promoters for the immediate-early genes *Arc* and *Fos* (*Reijmers et al., 2007*; *Wang et al., 2006*). We then used fluorescence-guided recordings (*Barth, 2007*) to evaluate cellular and synaptic changes specifically in the neural population activated during mating.

We focused first on interneurons, which were robustly labeled in Arc-GFP animals. Prior to recording, we assessed GFP labeling across the GC population with 2-photon microscopy. Naïve animals showed low levels of background GFP expression (*Figure 3A*), and 4 hr of sensory exposure to a male in the absence of mating produced little additional labeling over background (*Figure 3—figure supplement 1A*). However, mating combined with subsequent sensory exposure drove strong GFP expression in a subset of GCs, consistent with *Arc* immunolabeling in AOB in response to conspecifics (*Halem et al., 2001*; *Matsuoka et al., 2002*). We found robust labeling in both anterior and posterior AOB, in agreement with prior reports using histochemical staining (*Brennan et al., 1992*; *Halem et al., 2001*). Fluorescent activity reporters therefore identify mating-activated neural populations in live AOB tissue for targeted ex vivo electrophysiological measurements.

Using fluorescence-guided recordings of GFP-labeled GCs (*Figure 3B*), we asked whether the synaptic plasticity generated by mating was specific to these neurons. Unexpectedly, there was no difference between GFP(-) and GFP(+) populations of GCs for either amplitude or frequency of spontaneous excitatory input (*Figure 3C–D*; amplitude, 1.43 ± 0.15 vs 1.48 ± 0.13 mV; frequency, 7.13 ± 0.98 vs. 6.41 ± 0.84 Hz for unlabeled and labeled cells respectively; *Figure 3—source data 1*). Similarly, there was no significant relationship between the intensity of GFP expression and either amplitude or frequency of EPSPs (*Figure 3E–F*). These data suggest that mating globally increased synaptic drive onto inhibitory GCs without apparent specificity to the neurons activated by the stud male.

AOB output is relayed to behavioral and neuroendocrine centers by MCs, suggesting that memory specificity ultimately relies on changes in these neurons. Because MCs were only weakly labeled by Arc-GFP, we used an alternative Fos-GFP reporter line (*Reijmers and Mayford, 2009*). Fos-GFP levels were low in MCs from naïve females, but were robustly elevated in a subset of MCs after mating (*Figure 3G*; *Figure 3—figure supplement 1B*). Fos-GFP also provided more extensive labeling of GCs, suggesting that it captured similar sets of activated neurons, but at a lower threshold. As in Arc-GFP animals, we found no systematic differences in MC labeling in anterior vs. posterior AOB. In mated females, approximately 28% of detected MCs were classified as GFP(+) (intensity 4X greater than neuropil), but this is likely an overestimate due to difficulty in detecting unlabeled cells with live tissue imaging.

To correlate inhibitory plasticity with activation of MCs by the stud male, we repeated our measurements of GABAergic input using fluorescence-guided recordings (*Figure 3H*). This second dataset revealed a similar two-fold increase in mIPSC frequency in mated vs. naïve animals, with no change in amplitude (*Figure 3—figure supplement 1C–H*; frequency, 0.86 ± 0.14 vs.1.68 ± 0.17 Hz; amplitude, 58.02 ± 5.59 vs. 63.21 ± 5.53 pA for naïve vs. mated respectively). We evaluated the

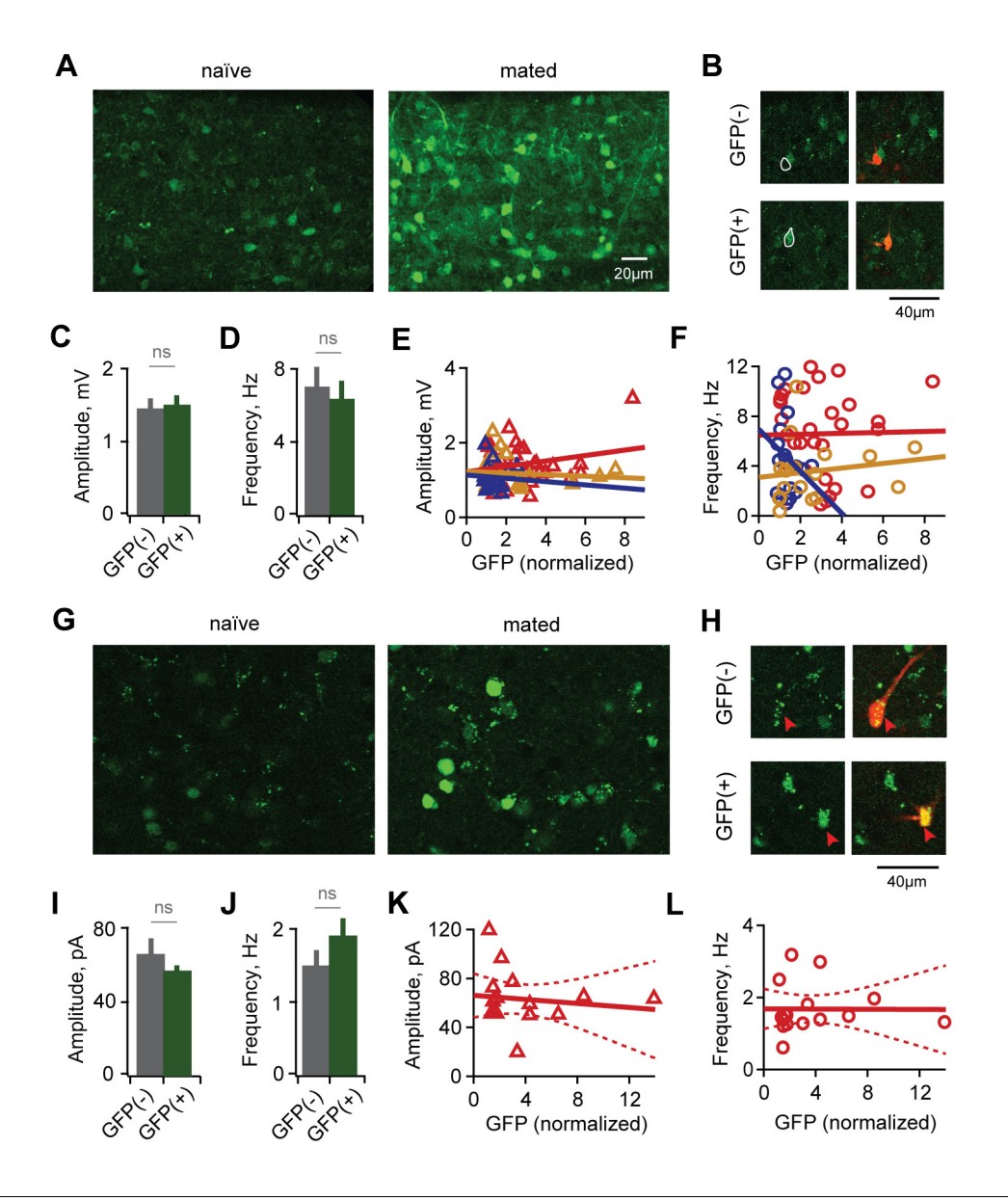

**Figure 3.** Synaptic plasticity is uncorrelated with activation during mating. (**A**) Arc-GFP labeling of AOB GCs activated by the stud male, visualized with live-tissue 2-photon imaging. Left, naïve control animal; right, mated female. (**B**) Fluorescence-targeted recordings of both unlabeled and labeled populations of GCs. (**C,D**) Mean amplitude and frequency of sEPSPs are similar for GFP(-) and GFP(+) GCs in mated mice (amplitude: p=0.82; frequency, p=0.58; t-test; n = 9 and 13 cells in 10 mice, GFP(+) and (-) groups subdivided from data in *Figure 3— figure supplement 1*). (**E,F**) GFP labeling is uncorrelated with either amplitude or frequency of spontaneous excitatory input to GCs (regression slope not different from zero; amplitude: p=0.70 and 0.22 for sensory-exposed and mated groups respectively; frequency: p=0.50 and 0.92; linear regression t-test; n = 17, 19 and 30 neurons in 5, 9, and 12 mice for naïve, exposure and mated groups). (**G**) Fos-GFP labeling reveals a subpopulation of mating-activated MCs (arrowheads). (**H**) Targeted recordings of stud-activated MCs. (**I,J**) Mean amplitude and frequency of mIPSCs are not significantly different between GFP(-) and GFP(+) MC populations (p=0.33 and 0.38 respectively; t-test; n = 8 and 5 cells in 5 mice; groups subdivided from data in *Figure 2*). (**K,L**) Amplitude and frequency of mIPSCs show no correlation with Fos-GFP intensity in MCs (regression slope not different from zero; p=0.64 and 0.97 respectively; linear regression t-test; n = 16 neurons from 5 mice). Dashed lines show 95% confidence intervals.

*Figure 3 continued on next page*

*Figure 3 continued*

The following source data and figure supplement are available for figure 3:

**Source data 1.** This spreadsheet contains the mean frequency and amplitude data for the individual neurons used to generate the bar plots shown in *Figures 3C, D, I and J*, comparing synaptic inputs to GFP(-) and GFP(+) neurons.

**Figure supplement 1.** Mating increases fluorescent labeling in AOB and increases inhibitory synaptic input onto MCs.

cellular specificity of synaptic changes within mated females by subdividing the second MC dataset into stud-activated GFP(+) neurons and a corresponding GFP(-) population. Similar to interneuron results, there was no significant difference in mean amplitude and frequency of mIPSCs between GFP(-) and GFP(+) MCs (*Figure 3I and J*; amplitude, 65.98 ± 8.15 vs. 57.86 ± 3.15 pA for unlabeled and labeled cells respectively; frequency, 1.43 ± 0.18 vs. 1.83 ± 0.31 Hz). Furthermore, there was no apparent correlation between GFP levels and properties of mIPSCs (*Figure 3K and L*). However, we cannot exclude the possibility that a larger sample may have revealed differences. Together, these results further indicate that imprinting drives synaptic plasticity in AOB inhibitory circuits. In contrast with established learning models, however, synaptic changes were widely distributed across both GC and MC populations with no apparent relationship to activation during mating.

## Mating enhances interneuron excitability

The lack of specificity in synaptic plasticity suggested that AOB output may be shaped by alternative mechanisms. One possibility is changes in intrinsic membrane excitability, which will alter the recruitment of AOB neurons by shifting the threshold for generating action potential firing. We tested for learning-induced changes in membrane excitability in AOB using graded current injections, focusing first on interneurons.

The responsiveness of GCs in Arc-GFP females was enhanced after mating, so that less current was needed to initiate firing, and higher firing rates were produced by the same current steps (*Figure 4A–B*). Increased excitability was also reflected in GC resting potentials, which were consistently depolarized in mated versus naïve animals (*Figure 4C*, −72.1 ± 1.6, −71.0 ± 1.6, and −66.3 ± 1.2 mV for naïve, exposed, and mated groups respectively). Other properties, such as membrane resistance and slope of the input-output firing function, were unchanged across groups (*Figure 4D*, $R_{input}$ = 469 ± 35 vs. 520 ± 34 for naïve and mated groups; p=0.31; ANOVA with post-hoc Tukey test), suggesting that increased GC responsiveness was largely determined by resting potential. Overall, mating increased the intrinsic excitability of AOB interneurons, suggesting that synaptic plasticity in inhibitory circuits is complemented by additional non-synaptic mechanisms.

To test whether changes in excitability were specific to mating-activated GCs, we examined the relationship between resting potential and Arc-GFP labeling. As with synaptic measurements, GC resting potential was uncorrelated with GFP intensity (*Figure 4E*). Furthermore, when we subdivided the GC dataset from mated animals into GFP(-) and GFP(+) populations, mean resting potential was similar for the two groups (*Figure 4F*; −67.0 ± 2.2 vs. −65.4 ± 1.9 mV respectively; *Figure 4—source data 1*). Together, the increased GC excitability after mating indicates that learning acts on intrinsic as well as synaptic properties of AOB neurons. However, intrinsic plasticity was widespread across interneurons, and lacked dependence on prior activation during mating.

## Mating attenuates MC responsiveness to repetitive stimuli

Because information about strain and individual identity is ultimately conveyed by MCs (*Arnson and Holy, 2013*; *Ben-Shaul et al., 2010*; *Luo et al., 2003*), the effects of learning should ultimately be reflected in their firing patterns. To further evaluate mating-dependent changes in AOB output, we compared responses to current injection in MCs from naïve and mated female Arc-GFP mice. In both groups, MCs responded to current injection with robust firing that decayed during the step (*Figure 5A*). In contrast to GCs, however, mating had no apparent effect on MC responses to a single stimulus. We found no difference in peak firing rate, total spike count during the train, or decay of firing between naïve and mated females (*Figure 5B–D*; peak firing: 15.0 ± 3.0 Hz vs. 17.4 ± 1.8

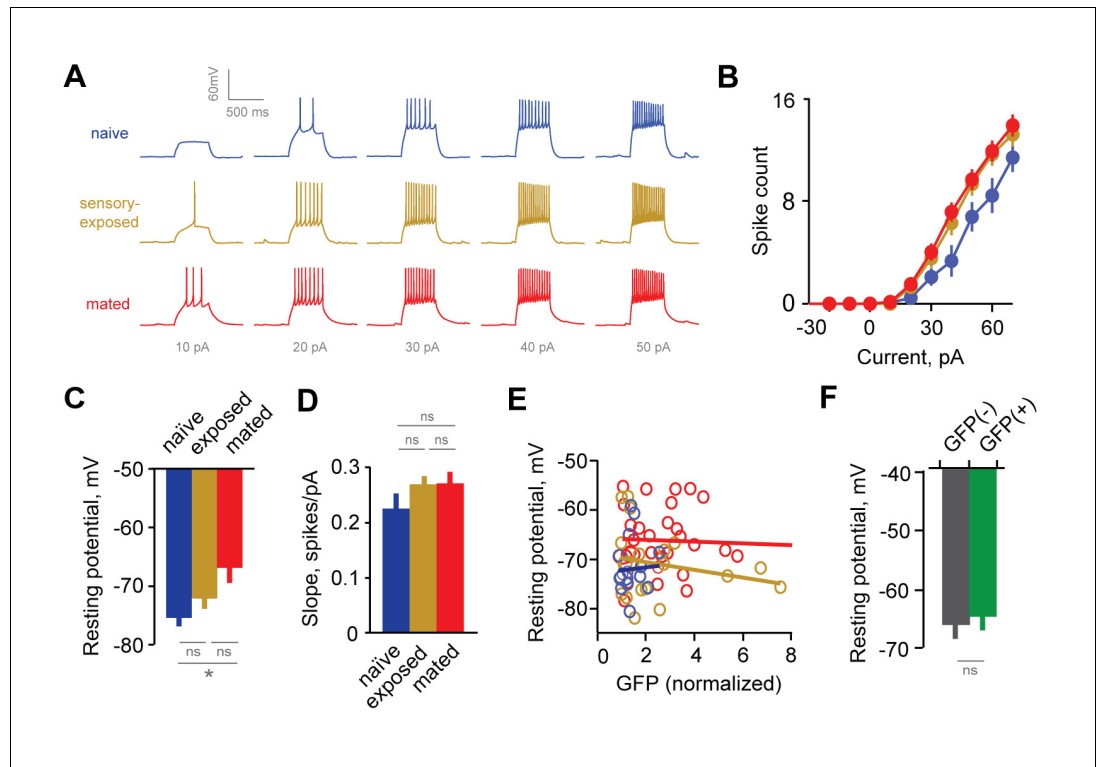

**Figure 4.** Experience alters intrinsic excitability of interneurons. (A) Representative responses to graded current injection for GCs from naïve, sensory-exposed and mated mice. (B) Current-firing plot shows a shift towards increased excitability of GCs from both mated and sensory-exposed females. (C) GC resting membrane potential was significantly hyperpolarized after mating (p=0.008 for mated vs. naïve; ANOVA with post-hoc Tukey test; F = 5.18; Fc = 3.14; n = 17, 19, and 31 cells from 5, 9, and 12 mice for naïve, sensory-exposed, and mated groups respectively). (D) The slope of the current-firing function was similar across groups (0.28 ± 0.01, 0.27 ± 0.02, and 0.28 ± 0.01; F = 0.14; Fc = 3.15; p=0.87; ANOVA with post-hoc Tukey test). (E) GC resting potential was uncorrelated with intensity of Arc-GFP labeling in both sensory-exposed and mated animals (slope not significantly different from zero; p=0.87, 0.37 and 0.81 for naïve, sensory-exposed and mated groups respectively; linear regression test; n = 17, 19 and 31 cells in 5, 9 and 12 mice). (F) In mated females, resting potential was indistinguishable between GFP(-) and GFP(+) GCs (−66.8 ± 2.19 vs. −65.4 ± 1.85 mV respectively; p=0.62, t-test; n = 10 and 13 cells in 10 mice, subdivided from the mated group in panel E). NS, not significant; *p<0.01.

The following source data is available for figure 4:

**Source data 1.** This spreadsheet contains the resting membrane potential and firing rate data for the individual neurons used to generate the bar plots shown in *Figures 4C, D and F*.

Hz; total spike count: 153 ± 28 vs. 176 ± 25 for naïve vs. mated respectively; *Figure 5—source data 1*). Despite robust increases in inhibitory input, therefore, MC firing for a single stimulus was unchanged.

Although immediate effects on MC output were not apparent, we further probed for changes over longer time periods. The hormonal changes that induce pregnancy block require prolonged AOB activity lasting several hours (*Li et al., 1994*; *Rosser et al., 1989*), timescales that typically encompass multiple sensory interactions (*Hull and Dominguez, 2007*). To approximate repeated activation of vomeronasal inputs (see *Figure 2—figure supplement 1*), we probed MCs with repetitive stimuli spanning several minutes (ten current injections, 20 s in duration, repeated every 60 s). Surprisingly, in mated females MC firing often declined dramatically across successive trials, so that even neurons initially responding with hundreds of action potentials ceased firing entirely (*Figure 5E*). While firing also declined in some neurons from naïve animals, MC attenuation was greatly enhanced after imprinting, so that average spike counts in mated females dropped to less

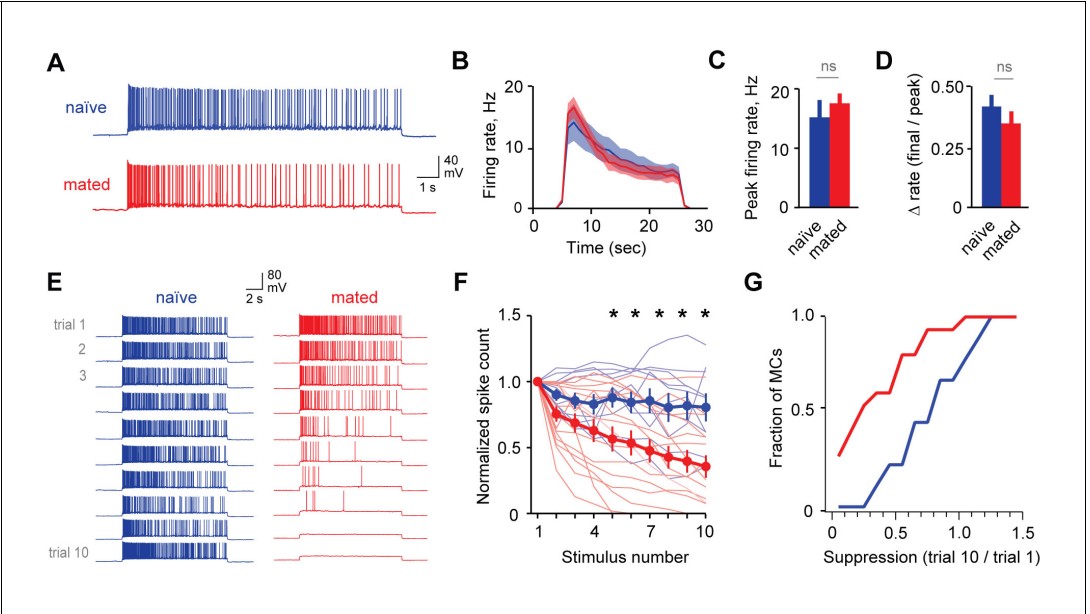

**Figure 5.** Mating reduces the responsiveness of MCs to repetitive inputs. (**A**) MC firing to an initial current stimulus is similar for naïve and mated females. (**B**) Firing rate profile averaged across all MCs from naïve and mated animals. (**C, D**) Mating has no effect initial MC output (peak firing rate: p=0.51; change in firing rate: p=0.55; t-test; n = 10 and 15 cells from 5 and 7 mice for naïve and mated groups respectively). (**E**) Example MC responses to repetitive stimulation. Firing is stable over time in naïve females, but drops dramatically over time after mating. (**F**) Average MC output across successive stimuli for naïve (blue) and mated females (red). Firing on 10th trial is 80 ± 11% (naïve) vs. 36 ± 8% of 1st trial (mated); F = 10.01, Fc = 4.30, p=0.003; ANOVA with post-hoc Tukey test. Light colors show individual neurons; dark traces show mean ± SEM. (**G**) Cumulative histogram showing increased attenuation in MCs from mated animals (p=0.027; Kolmogorov-Smirnov test; n = 9 and 15 neurons from 5 and 7 mice respectively). NS, not significant. *p<0.05; **p<0.01.

The following source data is available for figure 5:

**Source data 1.** This spreadsheet contains the firing rate and spike count data for mitral cells used to generate the bar plots and average data shown in *Figures 5C, D and F*.

than half than that of controls (*Figure 5F–G*; 80 ± 11% vs. 36 ± 8%). Together, our data indicate that mating leads to an unusual form of plasticity in MC membrane properties, where firing gains a striking dependence on recent history of activity. This metaplasticity in intrinsic excitability offers an alternative mechanism for attenuating AOB output, dramatically suppressing MC firing to repetitive stimuli and curtailing their responsiveness on timescales of minutes.

## Changes in excitability are specific to stud-activated MCs

To account for individual-specific recognition memories, AOB plasticity is predicted to be expressed selectively in the MC ensemble activated by the stud. Because targeted MC recordings were precluded by weak labeling in Arc-GFP mice, we tested for selective changes in excitability by collecting a second dataset using the Fos-GFP reporter. Within mated females, we compared responses of labeled MCs with unlabeled cells that presumably represent other, non-stud chemosignals. Mating drove robust increases in Fos-GFP relative to sensory-exposed controls, generating detectable labeling in approximately 25% of MCs (*Figure 3—figure supplement 1B*). After mating, both labeled and unlabeled MCs fired similarly to initial stimuli, consistent with responses in Arc-GFP mice (*Figure 6A–D*; peak firing rate, 15.2 ± 3.2 Hz vs. 16.2 ± 2.4 Hz; total spike count, 155 ± 39 vs. 133 ± 26 for GFP(-) and GFP(+) respectively; *Figure 6—source data 1*). We again used repetitive stimulation to probe for cell-specific plasticity in MC responsiveness. In unlabeled MCs, firing was stable across trials, or even increased slightly over time (*Figure 6E–F*; spike count of 10th vs. 1st trial, 115 ± 17%, peak firing rate, 87.0 ± 11.4%). In contrast, the output of GFP(+) MCs decreased markedly over successive stimuli (*Figure 6F–H*; spike count of 10th vs. 1st trial, 30.7 ± 12.3%; peak

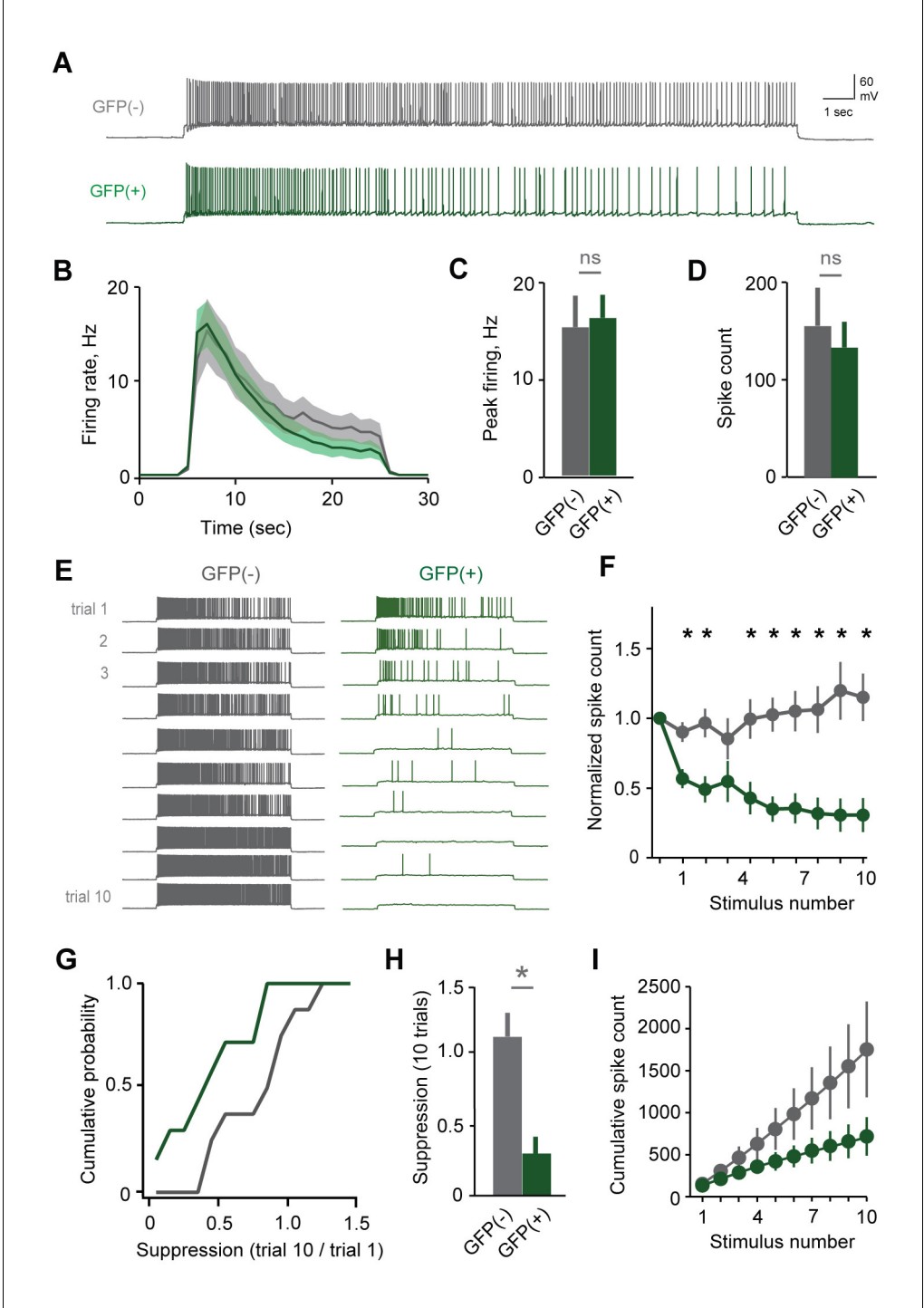

**Figure 6.** Plasticity in MC responsiveness is specific to mating-activated neurons. (**A**) Both GFP(-) and GFP(+) MCs show similar initial responses to current stimuli. (**B**) Mean firing rate profiles for GFP(+) and GFP(-) MCs in Fos-GFP females after mating (mean ± SEM; n = 7 and 11 cells in 5 and 7 mice). (**C, D**) Initial MC output is similar between GFP(-) and GFP(+) groups (firing rate, p=0.81; spike count, p=0.63; t-test). (**E**) Representative MC responses to repetitive stimulation in mated females. (**F**) After mating, GFP(-) MCs maintain consistent firing, but the output of stud-activated GFP(+) neurons is dramatically attenuated over time (mean ± SEM; n = 7 and 11 cells in 7 mice). (**G**) Cumulative histograms indicate a shift towards greater suppression in the MCs activated during mating (p=0.18; Kolmogorov-Smirnov test). (**H**) Mean suppression after 10 trials for GFP(-) and GFP(+) neurons (spike count on 10th

*Figure 6 continued on next page*

*Figure 6 continued*

vs. 1st trial: unlabeled, 115 ± 17%, p=0.41; labeled, 30.7 ± 12.3%; p=0.0017; t-test). (I) Cumulative action potential output of GFP(-) and GFP(+) MCs, averaged across all recorded neurons.

The following source data and figure supplements are available for figure 6:

**Source data 1.** This spreadsheet contains the firing rate and spike count data for mitral cells used to generate the bar plots and average data shown in *Figures 6C, D, F and H*.

**Figure supplement 1.** Correlated plasticity and GFP labeling are independent of selection criteria.

**Figure supplement 2.** Slow attenuation is absent in GFP(+) MCs labeled by sensory exposure alone.

firing, 29.4 ± 10.2%). Results were independent of criteria for selecting GFP(+) and GFP(-) populations (*Figure 6—figure supplement 1*). These data indicate that reduced excitability arises specifically in the MC ensemble activated by the stud during mating and subsequent sensory experience.

To ensure that MC excitability was altered by learning, rather than reflecting a pre-existing cell population in AOB, we performed parallel experiments where control females received sensory exposure to males without mating. This labeled a much smaller group of MCs, which responded with higher firing rates than unlabeled neurons in the same animals (*Figure 6—figure supplement 2A–C*). Membrane resistance was also higher in GFP(+) MCs (data not shown), suggesting that sensory stimulation alone preferentially recruits a group of high-excitability neurons similar to findings in neocortex (*Yassin et al., 2010*). In contrast to results in mated females, however, both GFP(-) and GFP(+) populations maintained consistent firing over all trials, despite their differences in initial responsiveness (*Figure 6—figure supplement 1D–F*). These data further indicate that dynamic changes in MC excitability result from imprinting, and emerge specifically in the population labeled during mating.

We estimated the net loss of output for stud-activated MC populations by plotting cumulative spike counts for matched groups of GFP(+) and GFP(-) neurons, which showed that total spike count diverged rapidly between the two groups (*Figure 6I*). Together, these results provide the first direct evidence of targeted, cell-specific changes in the AOB ensembles activated by conspecifics during social experience.

MC firing could potentially be shaped directly by changes in intrinsic membrane properties per se, or by prolonged synaptic inhibition that outlasts the stimulus. To distinguish between these possibilities, we tested MCs after blocking fast synaptic transmission with NBQX, APV, and bicuculline (10, 50, and 10 µM respectively). MC suppression was intact even after eliminating local circuit interactions, indicating that it did not depend on persistent inhibition (*Figure 7—figure supplement 1*). We further probed the source of reduced MC firing by examining membrane properties over the course of stimulus trains. Repeated stimulation led to a progressive hyperpolarization of MC membrane potential, both in randomly selected MCs in mated Arc-GFP females (*Figure 7A*) and in GFP(+) MCs in mated Fos-GFP mice (*Figure 7B*; *Figure 7—source data 1*). These changes were not predicted by initial MC resting potential, which was indistinguishable between naïve and mated females in Arc-GFP mice (*Figure 7C*; −58.0 ± 1.1 mV vs. −56.7 ± 0.7 mV respectively), and between GFP(+) and GFP(-) populations within mated Fos-GFP females (*Figure 7C*; −55.0 ± 1.5 vs. −53.9 ± 1.1 mV, p=0.56, t-test; n = 7 and 11 respectively). On average, GFP(+) MCs were hyperpolarized by −3.3 ± 0.5 mV vs. 0.2 ± 0.6 for GFP(-) neurons (*Figure 7D*; p<0.05, t-test). MC hyperpolarization was strongly correlated with loss of firing (*Figure 7E*). Hyperpolarization was accompanied by a slight reduction in membrane resistance (initial vs. final $R_{in}$, 486 ± 114 MΩ vs. 341 ± 27 MΩ; p=0.26, t-test). Together, these data further indicate that reduced MC output is due to changes in intrinsic membrane properties rather than altered synaptic inhibition, imparting stud-activated neurons with a sensitivity to recent firing that progressively dampens their output.

The hormonal changes that drive pregnancy block occur over several hours of sensory exposure, during which animals interact intermittently at varying intervals. To further probe the time course of plasticity, we probed randomly selected MCs in mated females using stimuli spaced 2 min and 3 min apart. Each firing bout led to hyperpolarization that decayed extremely slowly, lasting until the onset

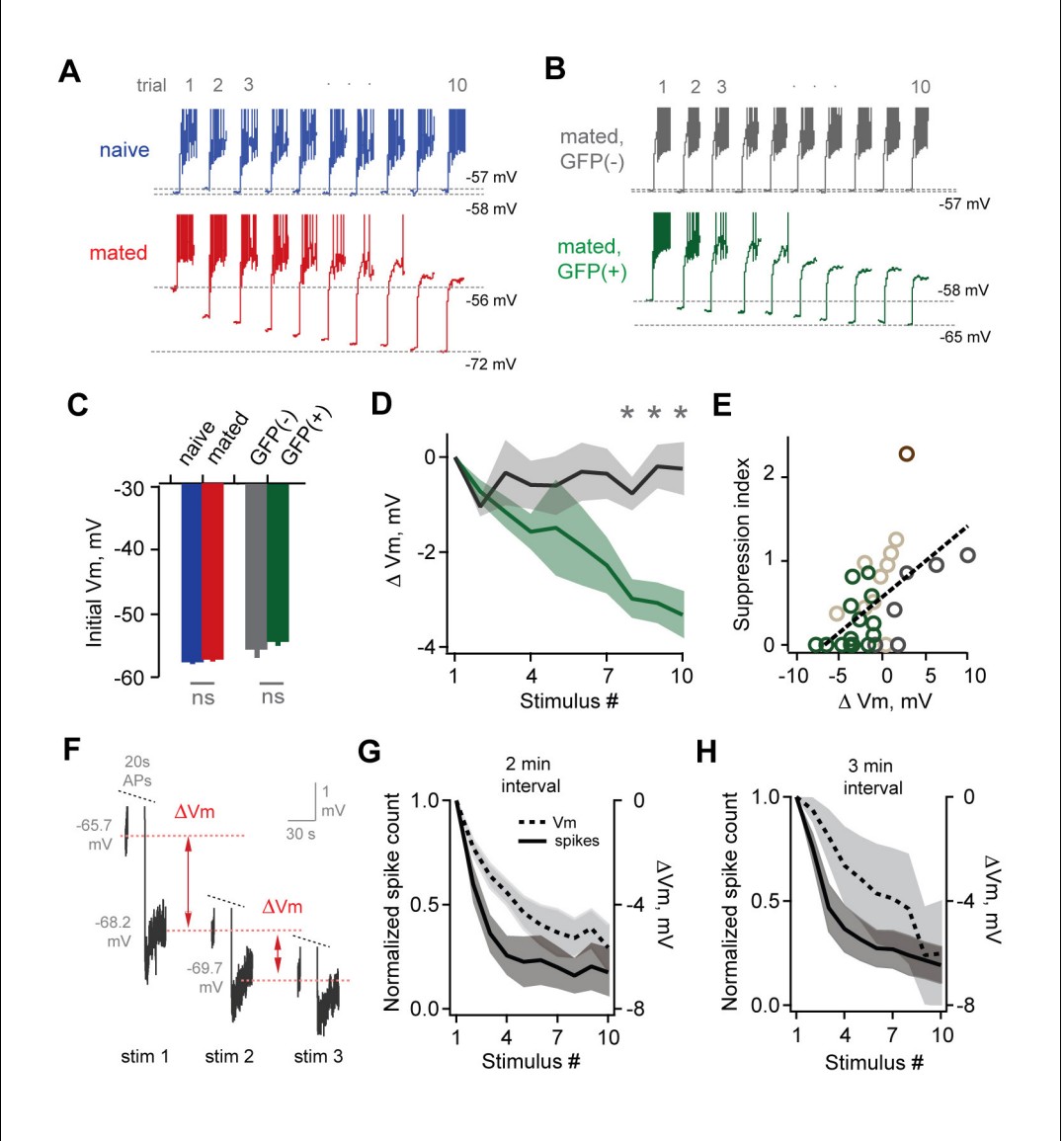

**Figure 7.** Loss of MC sensitivity results from progressive membrane potential hyperpolarization. (**A**) Representative MC responses to repetitive stimulation, showing initial resting potential and onset of firing for each trial. Progressive hyperpolarization was greatly enhanced in mated vs. naïve mice (red and blue respectively). (**B**) Within the AOB of mated females, hyperpolarization was selectively expressed in mating-activated GFP(+) MC populations. (**C**) Initial resting membrane potential was similar for MCs from naïve vs. mated females (blue and red respectively; p=0.76; n = 9 and 15 cells in 5 and 7 mice), and for labeled and unlabeled MC populations in mated animals (gray and green; p=0.56; t-test; n = 7 and 11 neurons in 5 and 7 mice). (**D**) Mean hyperpolarization during repetitive stimulation for GFP(+) and GFP(-) MCs (green and gray; upper and lower 1/3 of the recorded population; *, p<0.05; t-test). (**E**) Progressive loss of MC responsiveness is correlated with membrane hyperpolarization. Green, GFP(+); dark gray, GFP(-); light gray, intermediate. (**F**) Membrane hyperpolarization persists for > 2.5 min between stimuli. Traces show Vm before and after three successive spike trains delivered 3 min apart. Red dashes show step-like hyperpolarization lasting until the next stimulus. (**G,H**) Mean hyperpolarization and normalized change in firing for MCs tested with stimuli 2 min and 3 min apart.

The following source data and figure supplement are available for figure 7:

**Source data 1.** This spreadsheet contains the membrane potential data for mitral cells used to generate the bar plots and average data shown in **Figure 7C and D**.

*Figure 7 continued on next page*

*Figure 7 continued*

**Figure supplement 1.** MC hyperpolarization and attenuation of firing does not depend on persistent synaptic inhibition.

of the next stimulus (*Figure 7G*). Intervals of both 2 min and 3 min gave comparable levels of hyperpolarization and attenuated firing after 10 stimuli (*Figure 7G–H*; mean $\Delta$Vm, $-5.64 \pm 0.88$ and $-6.02 \pm 2.05$ mV for 2 and 3 min intervals respectively; normalized spike count, $0.176 \pm 0.116$ and $0.193 \pm 0.089$ for 2 and 3 min). When possible, we tested MCs again at 15 min and 35 min after the offset of stimulation, which revealed that suppression lasted for total periods of an hour or longer (normalized spike count, $0.149 \pm 0.080$ and $0.117 \pm 0.072$ at 15 and 35 min respectively; $\Delta$Vm, $-5.85 \pm 1.49$ and $-6.21 \pm 1.97$ at 15 and 35 min). These results indicate that MC plasticity persists on the timescales relevant to physiology and behavior in vivo.

## Discussion

Our data provide the first direct measurements of the synaptic and cellular effects of pheromonal learning in females after mating. We used ex vivo recordings to characterize changes in both excitatory and inhibitory neurons in AOB, and found broad enhancement of inhibitory circuits consistent with previous findings. However, using targeted measurements from identified mating-activated ensembles, we found that synaptic plasticity lacked the specificity thought to be required for recognition memories. Surprisingly, we also found a striking reduction in intrinsic excitability in MCs, suggesting a novel basis for storing sensory experience in AOB. Notably, this loss of responsiveness was confined to the set of MCs activated during mating, consistent with a stud-selective recognition memory. MC excitability also showed unusual dynamics, where output was initially unchanged, but instead progressively decreased depending on recent activity. Slow MC dynamics may selectively filter repetitive vomeronasal inputs over extended timescales, preserving sensitivity at the onset of behavioral encounters while reducing their longer-term impact on neuroendocrine centers that control hormonal status and pregnancy. Our data are broadly consistent with the selective MC plasticity proposed by existing learning models, but also suggest a new and unanticipated cellular mechanism that complements changes in synaptic strength.

### Inhibitory circuits in AOB

AOB inhibitory circuits had several unique properties. First, we found that MCs could strongly suppress their own firing via pronounced self-inhibition, consistent with robust GABAergic circuits in AOB (*Castro et al., 2007*; *Hendrickson et al., 2008*; *Shpak et al., 2012*). Self-inhibition in MOB MCs, in contrast, had much weaker effects on firing. Self-inhibition in AOB also emerged surprisingly slowly, building over several seconds so that it was strongest during prolonged MC firing on the timescales of natural sensory-evoked responses (*Ben-Shaul et al., 2010*; *Luo et al., 2003*). The basis for differences in self-inhibition in AOB and MOB are unclear, but may be linked to the prominent role of mGluRs in AOB (*Castro et al., 2007*).

Our data suggest that mating acts on multiple cell types and synaptic elements in AOB inhibitory circuits. Increased frequency of excitatory input to GCs implies upregulation of release sites in MC dendrites, and enhanced amplitude suggests strengthening of postsynaptic contacts, consistent with the ultrastructural enlargements seen in the postsynaptic density (*Ichikawa, 2003*; *Matsuoka et al., 2004*). Increased frequency of mIPSCs in MCs also suggests enhanced release of GABA from GCs, which could result either from changes in existing contacts or addition of new synapses via spine growth or recruitment of adult-born interneurons (*Mak and Weiss, 2010*; *Shingo et al., 2003*). Overall, however, this inhibitory plasticity appeared to have little effect on MC output driven by current injection. Increased release of GABA may be counterbalanced by short-term dynamics during extended firing (*Dietz and Murthy, 2005*), or alternatively changes in mIPSCs may reflect top-down inputs to AOB (*Fan and Luo, 2009*) that would contribute to spontaneous but not recurrent inhibition.

Memory formation has long been proposed to rely on inhibitory plasticity, selectively targeting the stud-activated MC population (*Brennan and Keverne, 1997*). In contrast, we found that synaptic

changes were widely distributed across both MCs and GCs with no observable dependence on activation during mating. While non-specific plasticity runs counter to existing models, our data do not exclude a role for inhibition in learning or sensory processing. GABAergic circuits may be differentially recruited in vivo, where complex natural cues activate larger MC populations (*Ben-Shaul et al., 2010*; *Meeks et al., 2010*). Inhibition strongly shapes MC firing in the intact brain (*Hendrickson et al., 2008*), and descending pathways targeting AOB are likely to further shape sensory responses (*Fan and Luo, 2009*). Overall, however, the nonspecific nature of inhibitory plasticity suggests it may act in concert with other, more targeted changes in the AOB circuit.

## Plasticity of intrinsic membrane excitability in MCs

Unexpectedly, imprinting had the most striking effects on intrinsic rather than synaptic properties, suggesting an alternative cellular mechanism for storing sensory experience in AOB. MC suppression was largely absent in naïve females, but was strongly increased by mating in two separate datasets, indicating that it is was generated de novo by learning rather than reflecting pre-existing AOB populations (*Yassin et al., 2010*). Experience-dependent changes in excitability often accompany synaptic modifications in both mammalian and invertebrate systems (*Daoudal and Debanne, 2003*; *Zhang and Linden, 2003*). Altered intrinsic properties support homeostatic regulation in cortical circuits, scaling cellular excitability to match long-term changes in sensory input (*Desai et al., 1999*; *Turrigiano, 2011*). Often, learning acts to enhance excitability (*Barkai and Saar, 2001*; *Zhang and Linden, 2003*), acting to amplify responses to trained sensory inputs (*Mozzachiodi and Byrne, 2010*), or to select neural populations encoding the learned cue (*Yiu et al., 2014*; *Zhou et al., 2009*). Here, in contrast, MC excitability was strongly reduced by pheromonal learning. This sign reversal is consistent with the fact that imprinting leads to the suppression of an otherwise default neuroendocrine response to sensory input. Notably, whereas other paradigms lead to static effects on excitability that are immediately apparent on testing, changes in MCs were dynamic and only emerged after strong activation. The excitability of control and 'imprinted' neurons was initially indistinguishable, and responses only diverged after cells had experienced substantial firing. MC hyperpolarization accumulated after each trial and lasted for at least 30 min after the offset of stimulation, suggesting that MCs display an unusual and highly integrative form of intrinsic membrane plasticity.

What are the potential advantages of intrinsic versus synaptic plasticity in AOB? Membrane excitability offers a simple way to selectively control the output of specific MC populations, whereas inhibitory plasticity would need to be coordinated across large sets of GCs, and further targeted to the specific synapses onto stud-encoding MCs. Intrinsic excitability may be particularly well suited to mediating learning in dedicated sensory pathways coupled to stereotyped behavioral responses. Interestingly, firing was similar for labeled and unlabeled MCs after mating, and only diverged after extended activity bouts. Thus, learning does not cause simple, static changes in MC excitability per se, but instead leads to metaplastic effects that impart sensitivity to recent firing levels. In metaplasticity of synaptic strength, experience shifts thresholds for potentiation and depression via changes in NMDA subunit composition (*Abraham, 2008*; *Lee et al., 2010*). Metaplasticity in the intrinsic excitability of MCs may rely on similar changes in composition of membrane conductances. Prolonged hyperpolarization and lowered membrane resistance are consistent with changes in potassium channels such as HCN or $Ca^{2+}$-dependent $K^+$ currents, which are present in main olfactory bulb and linked to learning in other systems (*Lin et al., 2008*; *Nolan et al., 2004, 2003*; *Stackman et al., 2002*; *Wang et al., 2007*). MC firing is also modulated by intrinsic conductances such as CAN currents, which boost synaptic responses (*Shpak et al., 2014, 2012*) and likely contributed to accelerated firing rates seen over the first few seconds of stimulation in our study. Interestingly, this current opposes the MC hyperpolarization we describe here, which appeared to dominate after more prolonged firing bouts and was most readily apparent only after mating. AOB MCs thus appear to express multiple activity-dependent conductances that dynamically modulate their firing depending on the strength, duration, and biological context of activity. The specific conductances responsible for slow MC attenuation remain to be established.

## Dynamic MC output and sensory representations

Dynamic, activity-dependent changes in excitability were a unique feature of AOB MCs. How may slowly adapting sensitivity contribute to sensory processing? One potential role is to high-pass filter

vomeronasal input, preserving responsiveness at the onset of social interactions for appropriate selection of aggressive, reproductive, or parental behaviors (*Burns-Cusato et al., 2004*; *Clancy et al., 1984*; *Stowers et al., 2002*; *Tachikawa et al., 2013*), while selectively attenuating the long-lasting, repetitive AOB activity required for the neuroendocrine changes that block pregnancy (*Li et al., 1994*; *Rosser et al., 1989*). Similarly, slow activity dynamics may help reduce interference between memories of different individuals. Different males have similar chemical signatures (*Harvey et al., 1989*), suggesting that they may also have overlapping neural representations in AOB. Eliminating the responses of stud-encoding MCs could therefore also disrupt representations of other, non-imprinted animals. Delayed changes in MC output may help minimize the impact of plasticity on overlapping sensory codes.

Fos-GFP labeling was transient, limiting our measurements to a period of several hours following mating. The effects of mating on pregnancy block, however, can last for many weeks (*Brennan et al., 1990*). It will be important to determine whether effects on MC excitability are maintained for similar time periods. Other, more permanent labeling strategies may allow plasticity to be tested at later time points (*Guenthner et al., 2013*; *Sakurai et al., 2016*). Alternatively, sensory memories could be stored initially in AOB and then transferred to other areas, as seen in other memory systems (*Preston and Eichenbaum, 2013*; *Ross and Eichenbaum, 2006*).

Mating is one of several biological contexts where animals show flexibility in vomeronasal-guided behaviors. Interestingly, many of these involve the loss of an otherwise default response, similar to the effects of mating. Males shift from attack to parental behaviors towards pups (*Tachikawa et al., 2013*; *Wu et al., 2014*), and regulate aggression towards other males to form dominance hierarchies (*Wang et al., 2014*). Behavioral responses to fear-inducing cues such as predator odors can also habituate with repeated presentation (*Takahashi et al., 2005*). It is currently unclear whether the behavioral plasticity seen in other paradigms relies on similar cellular mechanisms in AOB, and it will be important for future work to test this possibility.

Overall, our data reveal a novel form of cellular plasticity that emerges after mating in females, where slowly emerging, activity-dependent changes in intrinsic excitability dramatically attenuate the output of the MC ensemble activated by the stud male. It will be important for future work to test how this plasticity shapes sensory representations and neuroendocrine responses in behaving animals during social encounters. Changes in MC excitability could also contribute to flexibility in other vomeronasal-mediated behaviors, which often involve suppression of otherwise default sensory responses (*Tachikawa et al., 2013*). While the AOB is a critical node in the vomeronasal pathway, MC plasticity likely acts in parallel with broader changes across the extended network of brain regions that couple chemosensory input to behavior (*Dulac et al., 2014*; *Wu et al., 2014*).

## Materials and methods

### Mice

All experiments were performed in sexually mature adult female mice 8–14 weeks of age. Reporter lines were obtained from Jackson Laboratory (Arc-GFP, RRID:IMSR_JAX:007662; Fos-GFP, RRID:IMSR_JAX:018306) and bred in a C57Bl/6J background. Experimental Arc-GFP animals were heterozygous, maintaining a functional Arc allele. Animals were group housed in the Boston University animal care facility on a 12 hr light/dark cycle with *ad lib* access to food and water. Mice were anesthetized ≥5 days prior to experiment and received bilateral ovariectomies followed by implantation of estradiol capsules (*Bakker et al., 2003*). On the experimental day, estrus was induced with progesterone injection (16 µg/g), confirmed by vaginal smears and histological examination (*Caligioni, 2009*). At estrus onset, 3–4 hr after the beginning of the light cycle, females were paired with a sexually experienced male in their home cage for an additional 4 hr for mating and subsequent sensory exposure required for imprinting. Males typically attempted copulation within 20–30 min. Cases where males did not mount females were used as controls for sensory experience without mating. Sedated males were also used for sensory-exposure controls; activity-dependent labeling in the corresponding females was indistinguishable and these data were grouped together. Females in both mated and sensory-exposed groups were ovariectomized and progesterone-primed, while naïve females were unmanipulated. While mating success could not be evaluated in electrophysiology experiments, in a parallel group this protocol resulted in pregnancy in 9 of 11 gonadally intact

females. Reproductive encounters were video recorded and scored to quantify mating and behavioral interactions. All procedures were approved by the Boston University Institutional Animal Care and Use Committee and followed guidelines set by the US National Institutes of Health.

### Activity-dependent labeling

Activity-dependent labeling was visualized in each slice prior to electrophysiological recordings using a two-photon microscope (Prairie Ultima) with 920 nm excitation and a 20X NA 0.95 objective (Olympus), using consistent acquisition settings for laser power and detector gain across sessions. Immediately prior to establishing recordings, we acquired additional image stacks of GFP labeling for the field of view at each recording location (250 μm X 250 μm) using a 40X NA 0.8 objective (Olympus). Intensity was quantified for all detectable neurons using a circular region of interest centered on the soma. GFP intensity was continuously distributed, presumably reflecting graded levels of prior activity. Cells were classified as unlabeled or labeled using a threshold of $\leq 2$ or $\geq 4$ times background neuropil fluorescence respectively. For electrophysiological analysis, we performed similar analyses comparing the brightest third and dimmest 50%, 33%, and 25% of our recorded sample. Results were robust to classification threshold and comparison groups.

### Electrophysiology

Sagittal brain slices of AOB (300 μm thick) were prepared from female mice using a vibratome (VT1200S, Leica, Buffalo Grove IL). To preserve tissue health in adult animals, mice were deeply anesthetized with ketamine/xylazine and perfused transcardially with ice-cold modified artificial cerebrospinal fluid (ACSF) containing, in mM: 124 NaCl, 2.5 KCl, 1.25 $NaH_2PO_4$, 25 NaHCO3, 75 sucrose, 10 glucose, 1.3 ascorbic acid, 0.5 $CaCl_2$ and 7 $MgCl_2$. Slices were maintained using ACSF containing, in mM: 124 NaCl, 3 KCl, 1.25 NaH2PO4, 26 NaHCO3, 20 sucrose, 2 CaCl2 and 1.5 $MgCl_2$, continuously oxygenated with 95/5% O2/CO2. Slices were visualized with a two-photon microscope (Ultima, Prairie Technologies, Middleton WI) using a 40x water immersion objective and Dodt contrast imaging. Whole cell electrodes were pulled to tip resistances of 3–8 MΩ and contained the following internal solutions (in mM): current clamp, 135 K-gluconate, 2 $MgCl_2$, 10 HEPES, 0.4 EGTA, 2 MgATP, 0.5 $Na_3GTP$, 10 phosphocreatine disodium; voltage clamp, 115 CsCl, 25 TEA-Cl, 5 QX314-Cl, 0.2 EGTA, 4 MgATP, 0.3 $Na_3GTP$ and 10 phosphocreatine disodium. Alexa 594 was added to the internal solution to confirm cell identity in targeted recordings. Membrane voltage was not corrected for liquid junction potential. Electrophysiological data was collected at 29.5°C with a Multiclamp 700B amplifier (Molecular Devices, Sunnyvale, CA) and digitized at 10 kHz (National Instruments PCI-6321) using custom Matlab routines (Mathworks, Natick, MA). Action potential detection and analysis was performed using custom Matlab routines detecting zero-crossing membrane potentials. Changes in firing with repeated stimulation were quantified as a suppression index, calculated as the ratio of firing on the 10th vs. first trial. Synaptic currents and EPSPs were detected and analyzed in Igor Pro (WaveMetrics, Lake Oswego, Oregon) using Taro Tools (https://sites.google.com/site/tarotoolsmember/). Thresholds were chosen to maximize detection of synaptic events while excluding false positives due to recording noise. Thresholds were set at 10 pA for mIPSCs in MCs, and 0.25 mV for EPSPs in GCs. In both cases we estimate we detected at least 95–98% of events while limiting false positives to <1%, determined by visual inspection. GC results were consistent across a wide range of detection criteria. All chemicals were obtained from Sigma/Aldrich (NBQX), Tocris (BMI), and Alomone Labs (TTX). Receptor antagonists (APV, NBQX and Gabazine) were applied by bath perfusion. All results reported in the text and figures represent mean ± S.E.M.

### Statistical analysis

Statistical significance was calculated using t-test or ANOVA as appropriate, noted in results and figure legends. Distributions of miniature and spontaneous synaptic events were analyzed with the Kolmogorov-Smirnov test. Animals were randomly assigned to naïve, sensory exposure, or mating groups after recovery from surgery. Data collection and analysis were not blind to experimental conditions.

## Acknowledgements

We thank members of the Davison laboratory, Yoram Ben-Shaul, Tim Gardner, Adi Mizrahi, Jennifer Morgan, and Stephen Shea for comments and discussion. This work was supported by the Klingenstein Foundation, the Binational Science Foundation, and NIH/NIDCD (R21DC013894).

## Additional information

### Funding

| Funder | Grant reference number | Author |
| --- | --- | --- |
| National Institute on Deafness and Other Communication Disorders | DC013894 | Ian G Davison |
| The Esther A. and Joseph Klingenstein Fund | | Ian G Davison |
| United States - Israel Binational Science Foundation | 2015099 | Ian G Davison |
| United States - Israel Binational Science Foundation | 2013314 | Ian G Davison |

The funders had no role in study design, data collection and interpretation, or the decision to submit the work for publication.

### Author contributions

YG, Conceptualization, Formal analysis, Investigation, Methodology, Writing—original draft, Writing—review and editing; CB, ED, Investigation, Methodology; IGD, Conceptualization, Formal analysis, Supervision, Funding acquisition, Investigation, Methodology, Writing—original draft, Project administration, Writing—review and editing

### Author ORCIDs

Ian G Davison, http://orcid.org/0000-0003-0998-7676

### Ethics

Animal experimentation: This study was performed in strict accordance with the recommendations in the Guide for the Care and Use of Laboratory Animals of the National Institutes of Health. All of the animals were handled according to approved institutional animal care and use committee (IACUC) protocols (#14-034) of Boston University.

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
