## [Decision Letter]

Thank you for submitting your article "Neural mechanisms of social learning in the female mouse" for consideration by *eLife*. Your article has been reviewed by three peer reviewers, one of whom, Naoshige Uchida (Reviewer #1), is a member of our Board of Reviewing Editors and the evaluation has been overseen by Gary Westbrook as the Senior Editor. The following individuals involved in review of your submission have also agreed to reveal their identity: Tim Holy (Reviewer #2) and Joseph F Bergan (Reviewer #3).

The reviewers have discussed the reviews with one another and the Reviewing Editor has drafted this decision to help you prepare a revised submission.

Summary

Gao and colleagues examine neuronal plasticity in the context of "pregnancy block" (a.k.a. the "Bruce effect"), which involves a form of olfactory learning thought to be localized to the accessory olfactory bulb (AOB). A long-held view is that the critical change occurs at the synapses between granule cells (GCs) and mitral cells (MCs) in the AOB. Specifically, it has been hypothesized that mating causes a long-lasting increase in GC-to-MC inhibitory synapses in the AOB, and this suppresses the MC output transmitting the information about the mated male mouse. Although this hypothesis has been popular in the field, an important assumption -- whether this change occurs specifically in MCs signaling the mated male -- has not been tested experimentally.

The present study addresses this question by using transgenic mice (Arc-GFP and cFos-GFP) that allow for activity-dependent labeling of neurons with GFP. The authors first show that the frequency of mIPSPs increases in MCs in female mice that underwent mating, compared to naïve female mice or female mice that are exposed to a male mouse but did not mate. Importantly, the authors show that this change occurs similarly in mated and other control animals, and GFP-positive and -negative mitral cells did not differ in mIPSPs. The most significant finding is that the intrinsic excitability of MCs decreases in mated mice but not in naïve mice, as tested by their response to repeated, prolonged stimulations (10 repetitions of a 20-sec current injection). Furthermore, the authors show that this change occurs only in GFP-positive but not in GFP-negative MCs. Finally, they present evidence that the decrease in excitability occurs due to progressive membrane potential hyperpolarization of MCs across the repeated stimulations.

All the reviewers thought that the methods employed in this study are elegant and the results are highly significant. These findings provide a novel potential mechanism for the pregnancy block. Although all the reviewers were positive about the work, they raised important concerns or questions as described below. Most importantly, this study does not establish a causal link between the observed change in intrinsic excitability and behavior. To narrow this gap, additional information or experiments exploring some experimental parameters would be very informative. The essential points include some suggested experiments that may address these points. If the authors decide not to perform these experiments or the available data do not address these points, these issues should be discussed in the revised manuscript.

Essential points for revision

1) Is the change in MC intrinsic excitability specific to mating? It seems plausible that the change in MC intrinsic excitability is a general phenomenon not directly related to mating, except that mating is a behavioral stimulus that drives the AOB strongly and is therefore capable of revealing this mechanism. Do the authors have some data that addresses the question of whether other stimulus conditions (stimulation with pups or predators) also lead to similar changes or there is something special about mating?

2) The difference in MC excitability is revealed using "repetitive stimuli spanning several minutes (ten current injections, 20 sec in duration, repeated every 60 sec)". Can the authors justify this stimulation parameter (interval) based on the inter-bout interval between bouts of nasal contact during social encounters? Additionally, have the authors explored stimulation parameters that are effective or ineffective in causing the plasticity? How does the suppression depend on inter-trial interval? How does this compare to the interval between bouts of nasal contact during social encounters? This might involve some additional electrophysiology experiments and a more extensive and systematic analysis of video (as in Figure 2—figure supplement 1).

3) How long does the change in intrinsic excitability last and is it sufficient to explain the long-lasting effect at the behavioral level? Please comment on how the duration for which the change in intrinsic excitability lasts compare to the duration for which the effect can be observed at the behavioral level.

4) We would like to know whether the results in Figure 2 (as well as Figure 3—figure supplement 1) are sensitive to the choice of event criterion. First, it would be helpful if the authors explicitly explain their criteria for event detection, and indicate detected events in the figures. Second, please discuss whether the results are insensitive to a particular choice of event criterion.

5) Mitral cells were analyzed in Arc-GFP and cFos-GFP mice in different experiments. The authors mention that mitral cells were only weakly labeled by Arc-GFP mice. Is there evidence that Arc and cFos label a similar or overlapping population(s) of MCs? In other words, can the authors ensue that different experiments are looking at a similar set of MCs?

6) The authors compare between mated mice and mice with sensory experience without mating. The authors describe that "males typically attempted copulations within 20-30 minutes. Cases where males did not mount, females were used as control for sensory experience without mating". Can the authors provide evidence that other behaviors than mating, such as sensory explorations and locomotor activity, were similar between mated and non-mated mice? In particular, ensuring that the times spent for exploration of female mice are equivalent between the two groups appears to be very important. It would be helpful to compare the behavior of mice during the co-incubation period in a quantitative manner beyond Figure 2—figure supplement 1. It would be also helpful if the authors analyze the data after matching the time for exploration of female mice.

---

## [Author Response]

*Essential points for revision*

*1) Is the change in MC intrinsic excitability specific to mating? It seems plausible that the change in MC intrinsic excitability is a general phenomenon not directly related to mating, except that mating is a behavioral stimulus that drives the AOB strongly and is therefore capable of revealing this mechanism. Do the authors have some data that addresses the question of whether other stimulus conditions (stimulation with pups or predators) also lead to similar changes or there is something special about mating?*

As the reviewers note, our data do not address whether this form of plasticity is mating-specific. Plasticity could be unique to mating alone, or could reflect a cellular mechanism common to other forms of social learning, a possibility we favor. Alternatively, plasticity may simply be engaged by activity, which is presumably highly elevated during mating. These possibilities are not mutually exclusive. Excitability may change slowly in response to activity alone, but be altered more dramatically by highly arousing events like mating, perhaps facilitated by modulators such as noradrenaline that gate this form of learning. We have begun probing MC plasticity in other forms of social experience, and initial data suggest that other behavioral contexts also change MC excitability. However, MCs also appear to express multiple forms of plasticity in their intrinsic properties. Furthermore, different stimulus classes could have divergent effects depending on biological significance. For example, it may be advantageous to filter sensory input from social partners but not from predators. Addressing all of these points experimentally is impractical in the scope of this manuscript. We feel that these points are best addressed here by expanding the Discussion to explore these ideas.

*2) The difference in MC excitability is revealed using "repetitive stimuli spanning several minutes (ten current injections, 20 sec in duration, repeated every 60 sec)". Can the authors justify this stimulation parameter (interval) based on the inter-bout interval between bouts of nasal contact during social encounters? Additionally, have the authors explored stimulation parameters that are effective or ineffective in causing the plasticity? How does the suppression depend on inter-trial interval? How does this compare to the interval between bouts of nasal contact during social encounters? This might involve some additional electrophysiology experiments and a more extensive and systematic analysis of video (as in Figure 2—figure supplement 1).*

Intervals of 1 min were chosen on the basis of both behavioral observations and practical constraints on recording length. We have added a more extensive analysis of behavioral interactions in Figure 2—figure supplement 1 showing that, although widely distributed, the mean interval between bouts of investigation during pairing is indeed very close to 1 minute, and median values are even shorter (~15 sec). Thus, we believe that our in vitro paradigm is a reasonable approximation of physiological conditions. Behavioral investigation can be variable, and may also differ during the post-mating, pre-implantation period when pregnancy block typically occurs. To allow for reduced investigatory behaviors, we have also added substantial new data showing MC output is similarly attenuated by stimuli at longer intervals of 2 min and 3 min. Hyperpolarization and suppression are cumulative and appear to depend on the integral of past firing, so that low activity levels produce low amounts of suppression. Other than using low activity levels, however, we have not established parameters (such as long intervals, see below) that are ineffective at generating MC suppression.

*3) How long does the change in intrinsic excitability last and is it sufficient to explain the long-lasting effect at the behavioral level? Please comment on how the duration for which the change in intrinsic excitability lasts compare to the duration for which the effect can be observed at the behavioral level.*

Our new experiments also show that loss of MC sensitivity lasts for a period of >15-35 min after stimulation, indicating that MC responsiveness is regulated by integrating activity history over extremely long periods. This cellular mechanism thus has the potential to implement changes in AOB output on behaviorally relevant timescales of an hour or more.

The behavioral and physiological effects of mating persist for up to several weeks. Our Fos-GFP approach gives transient labeling that limits our measurements to a period of a few hours after sensory experience. We cannot currently address whether changes in MC responsiveness last beyond this period on the scale of days to weeks. More permanent labeling methods may allow testing of this idea in future work. Alternatively, our data reflect the initial stage of a dynamic memory process where information is stored at different sites at different time points. We have added a paragraph exploring these ideas in the Discussion.

*4) We would like to know whether the results in Figure 2 (as well as Figure 3—figure supplement 1) are sensitive to the choice of event criterion. First, it would be helpful if the authors explicitly explain their criteria for event detection, and indicate detected events in the figures. Second, please discuss whether the results are insensitive to a particular choice of event criterion.*

The Material and methods are expanded to include detection criteria. In GCs, we used a threshold of 0.25mV, which gave maximal sensitivity while avoiding false positives. In the revised manuscript we reanalyzed our GC data by adding an adaptive baseline, which further increased sensitivity. Thresholds were set initially by visual inspection of raw data and confirmed by extensive spot-checking. We estimate we detect 95-98% of events with negligible (<1%) false positives. This increased sensitivity gave small changes in quantification of GC EPSPs, which is updated in the text and in Figure 2 and Figure 3. We cannot lower thresholds further without introducing substantial noise contamination, but we have also repeated analyses using two additional, less sensitive thresholds (0.45 and 0.65 mV). Our conclusions are consistent across this wide range of detection criteria.

*5) Mitral cells were analyzed in Arc-GFP and cFos-GFP mice in different experiments. The authors mention that mitral cells were only weakly labeled by Arc-GFP mice. Is there evidence that Arc and cFos label a similar or overlapping population(s) of MCs? In other words, can the authors ensue that different experiments are looking at a similar set of MCs?*

We have added text to the Results section clarifying differences between the two reporters. We used Arc and Fos to label different cell types (granule and mitral cells respectively). We initially planned to use Arc-GFP for both, but after finding poor MC labeling with the Arc reporter we performed GC recordings in this line while importing the Fos reporter for MCs. Because Arc and Fos were used to label distinct cell types in separate experiments, our conclusions do not depend on the two reporters labeling the same populations. Fos gave more extensive labeling than Arc for both MCs and GCs in AOB, and for other brain regions as well, consistent with a lower activity threshold for Fos-GFP expression. The simplest interpretation is that Arc-GFP(+) cells are a subset of the Fos-GFP(+) population, consistent with studies showing overlapping expression of these immediate-early genes (refs). We have not directly tested overlap in AOB, so it remains possible that these reporters mark distinct populations. Again, however, our conclusions do not depend on this assumption.

*6) The authors compare between mated mice and mice with sensory experience without mating. The authors describe that "males typically attempted copulations within 20-30 minutes. Cases where males did not mount, females were used as control for sensory experience without mating". Can the authors provide evidence that other behaviors than mating, such as sensory explorations and locomotor activity, were similar between mated and non-mated mice? In particular, ensuring that the times spent for exploration of female mice are equivalent between the two groups appears to be very important. It would be helpful to compare the behavior of mice during the co-incubation period in a quantitative manner beyond Figure 2—figure supplement 1. It would be also helpful if the authors analyze the data after matching the time for exploration of female mice.*

We have performed further analysis of sensory investigation during the pairing period. Most females in sensory-exposed groups were paired with a sedated male. Females engaged in qualitatively similar behaviors, including intense investigation of facial and anogenital regions, huddling alongside the male, and general locomotion and exploratory behavior. While investigation was robust, it was performed at lower levels for sedated males relative to the mated group (e.g. lower number of investigatory bouts and total investigation time; see Figure 2—figure supplement 1), likely due to the lack of reciprocal behavior in the males. Thus, we cannot exclude the possibility that the MC plasticity we observe is due to differences in total sensory-evoked activity. In cases where males were alert but failed to mate, however, GFP labeling was similar to naïve and sedated-male groups despite investigatory behavior comparable to the mated group. Unfortunately in these cases we characterized GFP labeling but not electrophysiological effects, so we are unable to directly address the. We have added text to the Results section describing the differences in investigation times between groups. We note that this form of sensory learning is known to depend on noradrenaline release in AOB driven by vaginocervical stimulation.